# Pancreatic Organoids: A Frontier Method for Investigating Pancreatic-Related Diseases

**DOI:** 10.3390/ijms24044027

**Published:** 2023-02-16

**Authors:** Yuxiang Liu, Nianshuang Li, Yin Zhu

**Affiliations:** 1Department of Gastroenterology, The First Affiliated Hospital of Nanchang University, Nanchang 330209, China; 2Department of Gastroenterology, Digestive Disease Hospital, The First Affiliated Hospital of Nanchang University, Nanchang 330209, China; 3Jiangxi Institute of Digestive Disease, The First Affiliated Hospital of Nanchang University, Nanchang 330209, China

**Keywords:** pancreatic organoid, pancreatic cancer, personalized medicine, diabetes, pancreatic cystic fibrosis

## Abstract

The pancreas represents an important organ that has not been comprehensively studied in many fields. To fill this gap, many models have been generated, and traditional models have shown good performance in addressing pancreatic-related diseases, but are increasingly struggling to keep up with the need for further research due to ethical issues, genetic heterogeneity and difficult clinical translation. The new era calls for new and more reliable research models. Therefore, organoids have been proposed as a novel model for the evaluation of pancreatic-related diseases such as pancreatic malignancy, diabetes, and pancreatic cystic fibrosis. Compared with common traditional models, including 2D cell culture and gene editing mice, organoids derived from living humans or mice cause minimal harm to the donor, raise fewer ethical concerns, and reasonably address the claims of heterogeneity, which allows for the further development of pathogenesis studies and clinical trial analysis. In this review, we analyse studies on the use of pancreatic organoids in research on pancreatic-related diseases, discuss the advantages and disadvantages, and hypothesize future trends.

## 1. Introduction

The pancreas includes endocrine and exocrine glands that are responsible for the translation of macromolecular nutrients into absorbable fragments and the long-term sustained glycaemic balance. The pancreas cooperates with other organs to accomplish normal physiological functions, and dysregulation of any one of them can lead to dysfunction of the entire chain. Therefore, pancreatic disease not only causes local effects, but also results in multisystem disorders with homeostatic dysregulation that can be life-threatening. Hence, to better understand pancreatic-related diseases, numerous models have been applied in specific fields. There is no doubt that traditional models, including 2D cell culture, genetically engineered mouse models (GEMMs), and xenografts, have made tremendous contributions to disease investigation [1,2,3,4]. However, the translation of results into human clinical applications remains difficult [3,4]. After the abortion of TGN1412, the activator of CD28, further caution in the clinical translation of basic research is urgently needed [5]. The interspecies differences among in vitro cultured cell lines, mice, and humans may lead to the eventual failure of clinical translation. The two-dimensional cell culture is a prevailing model used by researchers ranging from advanced institutes to local laboratories. However, regarding individual pancreas express diverse genotypes, limited cell lines are not enough to recapitulate them [3]. Furthermore, genetic drift occurs in most cell lines after multiple passages, and would lead to unconvincing results [6]. In addition, the culture of these cells is not as simple as that of normal cells due to their microenvironment [7] and cell interactions [8]. GEMMs, which allow the introduction of target gene mutations into mice to simulate the human pancreas and perform experiments, are often limited by their repeatability in human models due to differences in gene diversity and metabolism [4]. Xenografts, including cell line-derived xenografts (CDXs) and patient-derived tumour xenografts (PDTXs), better mimic the cell environment [9], but establishing these lines is time- and capital-intensive, which makes the following disease progression difficult [2]. In summary, more feasible models are urgently needed for use in pancreas research.

Pancreatic organoids may provide a solution to increasing research demands. Since the first organoid created by Toshiro Sato et al. presented a differentiated cell type through the use of adult stem cells [10], organoid technology has inspired scientists worldwide to make great advancements in research. Many cell types have been confirmed to create organoids, including the stomach [11], intestine [10], colon [12], liver [13], prostate [14] and pancreas [15]. The first pancreatic organoids were generated in 2013. Huch et al. used mouse pancreas duct fragments to activate WNT signalling and express leucine-rich repeat containing the G protein-coupled receptor (LGR5), which produced a cyst-like structure that can self-replicate [15]. These cysts, or organoids, are still multipotent and can subsequently differentiate into specific cell types, such as acinar cells and duct cells. Pancreatic organoids are a promising model for translation to practical applications in drug screening and personalized medicine [16], because critical gene information is properly preserved even during gross replication [17]. In addition, the advantages of organoids include the relative harmlessness of extracting samples, the feasibility of applying them to humans, and the fact that subsequent disease modelling is not detrimental to health, which allow this technology to significantly reduce ethical compliance issues and increase the potential for organoid applications.

## 2. Creating Organoids

Organoid technology is a method for the 3D culture of cells extracted from the body to establish a functional organ in vitro. As early as 1944, researchers utilized anuran embryos to create pronephric ducts in vitro [18]. The history of pluripotent stem cell (PSC)-derived organ creation is extensive, and the publication of the usage of adult stem cells dates back to 2009, when Sato et al. innovatively transferred LGR5+ intestinal stem cells into unlimited proliferative cyst-like organs and differentiated them into several functional cell types [10]. However, the identification of organoids from spheroids is not simple, because under specific conditions, spheroids can also recapitulate architectural and functional aspects of the original tissue, which indicates the need for consensus regarding criteria for the identification of organoids [19]. Pancreatic organoids are notorious for their difficulty in establishing an extracellular matrix, particularly for pancreatic ductal adenocarcinoma (PDAC), in which the stroma is a key factor in tumour progression and drug applications [20]. In 2013, Huch et al. created mouse pancreatic organoids, and Boj then announced the creation of human and mouse PDAC organoids in 2015 [21]. These applications of pancreatic organoids have inspired additional research.

The sources of organoids can be different, such as embryonic stem cells (ESCs), adult stem cells (ASCs), tumour samples and pluripotent stem cells (PSCs), and different sources that recapitulate different characteristics. The methods through which different types of organoids are constructed vary. PSC organoid models require that extracted cells be first directionally induced to differentiate them by activin A pending endoderm formation before implantation into specific media. The ASC organoid model first requires the isolation of ASCs from a population of tissue-specific stem cells, which are then embedded in the extracellular cell matrix (ECM) containing preestablished combinations of tissue-specific growth factors. Tumour organoids acquire suitable tumour cells by mincing and digesting while being cultured and matured in a culture medium. Furthermore, the components in the medium for organoid establishment vary among species [22] and organs [23].

Pancreatic organoids serve as a novel research model and are combined with various other new techniques to achieve more satisfactory performance. CRISPR-Cas9 is a popular gene editing technology that enables the efficient knockdown and insertion of target fragments at particular loci. At the organoid level, this technology allows the editing of specific genes of primary cells in vitro, which in turn allows the generation of tumour-prone human cells and organoids [24]. Microfluidic chip technology integrates mechanical and biochemical external stimuli and precisely regulates local fluid perfusion, and its combination with organoid technology allows for the construction of an ideal in vitro platform for pancreatic cancer organoids [25] or islet organoids [26]. Bioprinting technology provides a new substrate or culture media for organoid modelling, which enables organoids to progressively rid themselves of the complex and numerous extracellular matrices at present and thus optimizes the experimental results [27]. A brief description of the construction of the pancreatic organoid is given below in Figure 1.

## 3. Applications of Organoids in Pancreatic Cancer

PDAC, which is a devastating disease with poor cure and recurrence rates, has risen in terms of the incidence of malignant tumours to 7th among men and 11th among women and to 6th in terms of the mortality rate related to malignant tumours in China, according to the 2021 statistics from the National Cancer Center of China [28]. PDAC is the most common type of pancreatic tumour, and accounts for 9% of all pancreatic tumours. Even when treated with developed therapy, few patients (less than 5%) survive longer than 5 years [29]. Possible reasons for this low survival rate include late diagnosis, rapid progression with early metastasis, and poor response to current chemotherapy [16]. These outcomes indicate that the recent research is far from satisfactory in providing a complete understanding of PDAC, and new methods should be applied to overcome the shortcomings of previous models.

Overall, PDAC is the most mature field of pancreatic organoids because preclinical research and clinical discovery have synchronously forged ahead. The related basic research includes investigations of the microenvironment and tumour initiation, whereas the established model has been applied to build a biobank for drug screening and personalized medicine. Currently, the main research areas related to pancreatic cancer organoids include the establishment of an organoid model of PDAC, the construction of models from normal cells using CRISPR-Cas9 technology, and the establishment of an organoid biobank for precision medicine for pancreatic cancer [30]. With respect to pancreatic cancer organoids, we will address this topic from the perspectives of cancer initiation and progression and discuss directions for future clinical applications. The overall application of organoids shown below in Figure 2.

### 3.1. Cancer Initiation

Pancreatic cancer is characterized by KRAS mutations that universally occur in over 90% of cancers during the initiation period [31]. In addition, pancreatic tumours usually represent the fruit of cooperation because TP53, CDKN2A and SMAD4 mutations also play a significant role in cancer initiation [32]. The overexpression or inhibition of regulators and effectors of KRAS and its downstream signalling pathways accelerates tumour progression, but fortunately, this process is influenced by multiple factors and often occurs at the early stages of tumour formation [33]. KRAS mutation initiates the tumour process as pancreatic intraepithelial neoplasia (panIN) with a relatively long latency or other tumour-suppressing gene mutations, and panIN progresses to pancreatic cancer [34].

KRAS encodes a member of the Ras family of small GTPases, and two of its paralogues also exhibit a high potential for gene mutation [35]. Prior to the applications of organoids, GEMMs carrying mutated KRAS genes from embryos were created and exposed to other key gene mutations to investigate PDAC. To date, organoids have been applied to establish various tumorigenesis models [36,37,38], but few researchers waited for single KRAS-induced tumorigenesis because they used gene editing [37] or multiple gene mutations [38] to accelerate progress. The creation of organoids is relatively specific, and pancreatic organoids from mouse primary cells implanted in other mice have failed to achieve a convincing result because tumorigenesis may be considered artificial [34]. In addition, because KRAS contains multiple alleles, distinct alleles can alter tumour characteristics. For example, under the same EGFR therapies, different KRAS mutations may yield different outcomes, and KRAS^G12C^ shows less sensitivity than KRAS^G13D^ [39]. These results partly clarify the heterogeneity resulting from the same therapy among different pancreatic tumour organoids.

Although tumorigenesis results from multiple factors, many other genes or factors also control the deterioration progress. BCAT2, as a key enzyme for lipid digestion, can accelerate the uptake of branched-chain amino acids (BCAAs) and enhance PanIN formation in an organoid model in a dose-dependent manner [40]. Moreover, the convergence of YAP/TAZ is critical in the early reprogramming of acinar cells and can be manipulated to create organoids [41]. Loss of Lats1/2 may specifically activate YAP/TAZ and then lead to the rapid progression of dysplastic lesions in organoids [42]. Interestingly, although the potential mechanism linking these remains unknown, obesity has also been recapitulated in a mouse organoid model at different disease stages, demonstrating a supporting tendency from tumour growth to metastasis at all phases [43].

### 3.2. Microenvironment

The tumour microenvironment is universally recognized as a crucial factor for tumour initiation and central tumour targeting [44]. In addition, the extracellular matrix of PDAC, which is known as a mixture of numerous complicated compartments interacting to form a highly dense mass with an inefficient, leaky vascular network, is considered an important factor responsible for a low perfusion rate, which in turn causes a suboptimal microenvironment for drug delivery [45,46,47,48]. Recent studies on PDAC organoids have mainly relied on human tumour tissue or cell line-derived organoids cocultured with cancer-associated fibroblasts (CAFs) from patients. Notably, even when used in the same cell lines and cocultured with fibroblasts, slight differences in size and structure are detected between organoids [46]. Human-derived organoids are favoured because they depict a similar organ with cell polarity, multiple cell divisions, and specific growth factor-dependent proliferation [49].

CAFs play a key role in tumour initiation and progression, but individual CAFs extracted from different donors show diverse expression. When cocultured with organoids, a divergence in CAFs may be observed among different organoids. CAFs form the main portion of the dense stroma, which is also a shelter of tumour cells that reproduce independently while prohibiting chemotherapy and targeted drugs from specialized elimination [47]. The extracellular matrix constitutes more than 90% of the tumour volume and interstitial cells, including myofibroblasts, CAFs, pancreatic stellate cells, and myeloid-derived suppressor cells. These cells interact with each other to create a stable stroma around the tumour mass [50]. The overlap of the extracellular matrix secreted by CAFs oppresses the tumour substance and induces tumour hypertension, resulting in the suppression of blood vessels inside the tumour and ultimately insufficient drug delivery [45,46,47,48]. Because fibroblasts create dense stroma-connected tissue around the mass, the PDAC organ is considered stiffer than the normal pancreas, and prohibits blood perfusion and drug delivery. CAFs are created when they are cocultured with mouse PDAC organoids, and through lysyl oxidase-dependent crosslinking, CAFs can rebuild the fibrotic environment [51]. In addition, unlike tumour cells, which are generally thought to activate CAFs [52], a desmoplastic matrix is activated; hence, exosomes are oversecreted, resulting in enhanced resistance to chemotherapy [53], and a suboptimal effect of gemcitabine in the organoid model [51]. In an organoid model, the sub-microenvironment is characterized by fibrotic plasticity [54]: a model with a microenvironment rich in coordinated fibroblasts will exhibit tumour microenvironment heterogeneity during tumour progression, whereas that with a microenvironment with fewer activated fibroblasts supports cell differentiation [55]. Due to the presence of different subtypes of CAFs, different signalling pathways implicate two contradictory phenotypes, including inflammatory fibroblasts and myofibroblasts [56]. Controlling the culture conditions in organoids can selectively create a specific microenvironment and yield different tumour results [52]. Hence, the traditional notion of a generally tumour-supporting microenvironment needs reconsideration. Additionally, by controlling the abundance of paired related homeobox 1 (Prrx1), the in vivo model shows that it will impact the mesenchymal plasticity of CAFs and influence gemcitabine [57].

Moreover, although many other factors may not be involved in tumorigenesis, their abnormal presence is associated with a malignant intensity in tumour progression and indicates a potential for new therapeutic targets. Metabolic interactions have been modelled to investigate microenvironmental changes and thus determine the optimal conditions for tumour progression. Sterol O-acyltransferase 1 (SOAT1) responds to the mevalonate pathway to acquire cholesterol, which has been found to participate in all processes in tumour progression in an organoid model with differential regulation of cholesterol homeostasis [58]. In addition, PDAC organoids exhibit central hypoxia and require hypoxia inducible factor-1 (HIF-1) to control glycolysis. Genetic disruption or pharmacological inhibition of ubiquitin-specific peptidase 25 (USP25) will impair the glycolytic process by inhibiting the transcriptional activity of HIF-1 and significantly inhibit the growth of organoids [59].

Except for CAFs, many cells in the matrix have also been discussed based on the use of cocultures to explore their function in the PDAC environment. For example, pancreatic stellate cells are activated by a wide variety of external stimuli to become myofibroblast-like cells, produce metabolic changes, and promote cancer [60]. Coculturing with pancreatic tissue cells rapidly enhances cell viability, migration, and tumour cell invasion. N-acetyl-cysteine strongly inhibits pancreatic stellate cells but not tumour cells, resulting in the suppression of tumour growth and liver metastasis in KPC mice [61]. In addition, coculture technology enables the in vitro study of pancreatic cancer-related stromal and immune interactions using organoids, and multiple compartments can interact to yield a better model for even more profound research. Tsai published the first report of a coculture of pancreatic tumour cells, matrix and immune cells in an in vitro 3D model and found the increased activation of myofibroblasts, such as CAFs, and the infiltration of tumour-dependent lymphocytes [49].

Although immune therapy, particularly the immune checkpoint blockade, has been successfully applied to treat numerous malignant tumours, it has not shown much benefit against PDAC because it is still less optimal in most pancreatic cancers [1,62]. Compared with other tumours, pancreatic cancer infiltrates contain fewer T cells, which also accounts for the poor cure rate. By coculturing with organoids, tumour-suppressing immune cells are even more closely recruited around tumour organoids, which can increase the function of these cells and serve as potential therapies. Immune-suppressing microenvironments result from not only the deactivation of immune cells but also a comprehensive effect of interactions among CAFs, immune cells and other matrix cells [63]. CAFs not only directly affect the activity of immune cells, but also indirectly lead to the dysfunction of immune effector cells by upregulating the expression of immune checkpoint molecules on the cell surface and reshaping the ECM in the microenvironment. Immune cells cocultured with human or murine cancer organoid models exhibit a downregulation of T-cell effector function [64]. Bishehsari et al. innovatively investigated the interaction between tumour cells and macrophages [65]. By analysing the phenotypic change in macrophages exposed to KRAS mutation, these researchers reported the induction of KRAS to shift macrophages towards the tumour-supporting subtype and accelerate the cancerous phenotype in the epithelium [65].

### 3.3. Drug Screening and Personalized Medicine

PDAC is the most common and malignant tumour in pancreatic cancer. At present, surgery is the most beneficial therapy but remains limited to the early stage of PDAC and is indicated for only a few of the many people who experience recurrence. Currently, gemcitabine/nab-paclitaxel or FOLFIRINOX is popular for most patients without surgical indications, and yields satisfying results at an early stage [66]. With a complete and systemic regimen of neoadjuvant chemotherapy (NAT), even some unresectable tumours can be transformed into operable tumours with an evident response, which may improve the cure rate of patients and prolong their lifespan [67]. However, the long-term survival remains limited to less than one year [68]. Hence, to discover better clinical applications, related research involves several fields with variable models. Due to difficulties in the translation of laboratory results into clinical application, mouse models and 2D cell culture are becoming problematic. Pancreatic organoids, which are capable of recapitulating the normal physiological environment, have received increasing attention. In addition, organoids can be established in patients who are chemotherapy-naïve or post-NAT and are in accordance with the resected tissue. Because the time needed to construct organoids is not long, a longitudinal study of pancreatic cancer can be performed to discuss their drug sensitivity and tumour evolution.

Similar to other models, organoid models are also challenged by their lack of translatability to clinical situations. To demonstrate the feasibility of the translation of organoids from the laboratory to the clinical environment, Seppälä et al. developed a related prospective clinical trial. These researchers demonstrated the existence of a meaningful period for the generation of data from organoids that can facilitate precise cancer treatment and further emphasize that the sensitivity of the ex vivo model can prospectively predict the chemotherapy response based on biomarker comparisons [67]. Additionally, although pancreatic cancer exhibits different morbidity and mortality rates between people, human diversity is also an important factor in chemotherapy response. Demyan et al. published the most culturally diverse organoid biobank with the largest population for PDAC, and demonstrated that the diversity of organoids varies from resection to biopsy, even after chemotherapy [69].

To identify the chemical resistance mechanism of PDAC, organoids also deliver a reliable approach. Gemcitabine and FOLFIRINOX are well-known chemotherapy regimens in PDAC, but rapid resistance after dosage largely prevents individuals from completing the entire course of medication, and this finding warrants further research [70]. SMAD4 is a tumour-suppressing factor in the TGFβ signalling pathway. The loss of SMAD4 may cause cell mitosis and increase sensitivity to cell cycle-targeted drugs. The upregulation of cell cycle-related genes such as CDK1 has been observed in SMAD4-deleted PDAC and is associated with increased resistance to gemcitabine and other cell cycle-targeting agents [71]. A hypoxia-convergent gene, RETSAT, was recently defined as a factor that helps cancer cells adapt to a hypoxic microenvironment. Novel research has concluded that this replication fork protein promotes fork restarting and upregulates gemcitabine resistance [72]. FOLFIRINOX is an important component of adjuvant chemotherapy in advanced cancer and improves overall survival compared with that observed with gemcitabine monotherapy [73]. Although the essential factors for individual drug resistance still require in-depth investigation, organoids from patients treated with the neoadjuvant FOLFIRINOX provide a fascinating tool for predicting the responsiveness to therapeutic drugs and analysing treatment failure [74]. Farshadi et al. constructed a database of 10 primary and FOLFIRINOX-treated organoids and clarified that the histological, genetic, and transcriptional characteristics of the organoids recapitulate those of primary tissue. The organoids were then treated with single agents or with a combined regimen of FOLFIRINOX, and the comparison clearly showed the existence of significant drug resistance in tumour tissues after FOLFIRINOX treatment.

Despite all the research published to date, a sufficiently curative target for pancreatic cancer in clinical settings is still lacking. Recent mutation drivers capable of targeting factors include PD-1, BRCA2 and uncommon KRAS^G12C^ mutations [75]. Predictive biomarkers are still lacking and are urgently needed for precision medicine. In this context, organoids are a promising model for identifying potential targets, diagnoses, and novel therapies for PDAC. Due to the increasing awareness presented, the subtype of PDAC has become even more notable and validated as a novel research direction. Despite the many classifications for PDAC identified, two major subtypes have been established: one is characterized by the expression of basal markers such as cytokeratins, indicating a poor prognosis, and the other is characterized by ductal markers such as GATA6 and predicts a relatively better result [75,76]. Thus, organoids from these tumour tissues yield different results following chemotherapy [75]. Shi et al. reported that the whole-genome sequencing of cancer organoids clarifies that not only coding protein mutations but also noncoding gene abnormalities may cause cancer occurrence and metastasis in the development of pancreatic cancer [77]. PAF1 is defined as a transcription factor complex for the maintenance of pancreatic tumour stem cells, and poorly differentiated cancer exhibits higher upregulation [78]. Saswati et al. demonstrated a PAF1-PHF5A-DDX3 subcomplex that endows colocalization and a subcomplex level increase among tumour specimens, human-derived organoids and mouse-derived organoids. The depletion of this complex downregulates cancer stem cell-related genes and transcription factors that regulate pathways such as JAK/STAT and ERK/MAPK, which ultimately leaves tumour cells with a reduced capacity for self-renewal [79]. Intracellular blood components are also being included in the discussion of pancreatic cancer development. Following preclinical and clinical research, the influence of tumour cell-induced platelet aggregation on tumour progression has become increasingly evident. Although pancreatic organoids can maintain cell polarity and interact with the extracellular matrix, fully understanding the crosstalk between pancreatic cancer and platelets offers a prospective direction for lateral diagnosis and treatment [80]. Powerful immunotherapies are not sufficient for overcoming devastating pancreatic cancer [81] but do have advantages. Organoid models verify that CD8+ immune cells have higher apoptosis activity in cancer cells; thus, immunotherapy synergizes with Pin1-targeted drugs to eradicate advanced PDAC [82]. Pancreatic cancer remains an actively discussed field in pancreatic organoid research.

Pancreatic cancer remains the most favourable cancer in pancreatic organoid research. Due to their excellent imitation ability and ethical feasibility, organoids are distinct from many research models. Pancreatic organoids have a wide range of applications in tumour occurrence and development and precision medicine. Although organoids are aggregates of cells and lack a surrounding environment [83], coculture technology enables a better understanding of the interaction of organoids in the tumour microenvironment. Additionally, as organoid technology continues to develop, the culture patterns of organoids are also changing. As the most prevalent organoid culture medium, Matrigel has facilitated the construction and refinement of organoids. Matrigel was derived from mouse sarcoma and has over 1800 components, the effects of which have not been fully elucidated. In addition, the variation between batches and potential immunogenicity have also affected its wider application. Hence, Matrigel-free extracellular matrices, such as hydrogels and recombinant protein gels, which allow independent changes in the chemical composition within the matrix, are gradually expanding the scope of application [27]. Organoids-on-a-chip technology provides a better blood supply and tissue environment for organoids and overcomes the limitation of a lack of blood supply caused by organoid angiogenesis disorders, and an improved chip technology may provide the possibility of high-throughput drug screening [84]. It is foreseeable that with the gradual maturation of organoid technology, drug screening and precision medicine related to pancreatic cancer will be further improved, and targeted therapy for new targets may significantly improve the curative effect of pancreatic cancer treatment.

## 4. Diabetes

Diabetes is caused by multifactorial contributions to poor glycaemic control, and can ultimately lead to multiple organ dysfunction. Type 1 diabetes represents an absolute insufficiency of insulin, whereas type 2 represents a relative insufficiency. Islet organoids, representing a novel islet replacement therapeutic modality, hold great potential due to their unique adaptability and long-term sustainability. Islet organoids are relatively different from pancreatic organoids. Organoids used for pancreatic cancer research are often ductal epithelial cell resources, whereas pancreatic islets with endocrine function are frequently used in the study of diabetes β cells to establish organoids.

There remain many unknowns in diabetes research that need to be elucidated. Many factors, such as interspecies differences, ethical controversy and donor deficiency, hinder diabetes research to a certain extent and thus delay the development of corresponding diagnostic and treatment modalities [85]. Currently, islet transplantation is the most effective treatment at the late stages of both type 1 and type 2 diabetes [86], and the overall survival expectation of direct islet transplantation is similar to that of whole pancreas transplantation [87]. However, as immunotherapy has become more controlled, the continuous promotion of islet transplantation is also becoming more mature, which eventually also leads to a substantial increase in the demand for a donor pancreas. Hence, islet organoids, which are generated by the in vitro differentiation and aggregation of embryonic or adult stem cells, are under continuous discussion as a possible regenerative medicine technology. The most ideal transplantation method is autologous transplantation, which can minimize autoimmune damage; however, the high conversion cost is difficult to ignore [88]. The construction of organoids with low or no cellular immunity thus becomes an active research direction. There are currently multiple ways to address this issue. Eiji et al. provided a protocol to create human islet organoids and, by overexpressing immune checkpoint proteins, observed their abilities to provide glycaemic control in immunocompetent diabetic mice and escape from possible cellular immunity [89]. In addition, human amniotic epithelial cells (hAECs) are considered to possess regenerative, immunomodulatory and anti-inflammatory properties [90]. The integration of hAECs into organoid models not only enhances the blood supply but also elevates the insulin secretory function, modulates immune responses, reduces posttransplant inflammatory damage, and prolongs islet survival and transplant success [91,92].

Although islet organoids are currently far from clinical application, many relevant studies have demonstrated their feasibility and optimized their formulations to achieve more effective therapeutic outcomes. Prevascularisation is a commonly used method to promote the maturation of pancreatic islet organoids. Human umbilical vein endothelial cells and hAECs are used to enhance cell vitality and support the formation of new blood vessels [93]. Gene manipulation has also been proposed to enhance islet maturation following the inducible knockdown of LIN28B in the pancreas. The expression of many markers of islet B-cell maturation and function, including PDX1, NKX6.1, MAFA, PAX6, and ISL1, is elevated [94].

Regardless of the issue of immunomodulation, the clinical translation of organoids is limited by a relative lack of understanding of their structure versus function and the current difficulty in meeting the needs of long-term glucose maintenance. As a highly anticipated application, islet organoids are regarded as an unlimited organ resource for transplantation, and diabetes therapy is being integrated with technologies such as gene chip and 3D bioprinting to enable researchers to obtain a deeper understanding of diabetes [95].

## 5. Other Applications

Pancreatic ductal epithelial cells are similarly an important component of the exocrine system of the pancreas. The pancreatic juice released by ductal cells not only drives the transfer of pancreatic enzymes to the gastrointestinal tract, but also delivers an alkaline environment that limits further pancreatic enzyme activation within the pancreas [96,97]. The construction of suitable organoids that can overcome the current shortcomings of commonly used models to better study ductal epithelial ion transport may facilitate the study of pancreatitis and pancreatic cystic fibrosis [96]. Recent superficial evidence shows that compared with isolated pancreatic ductal tissue, pancreatic organoids grown in vitro are equally capable of exhibiting similar gene expression profiles and are thus functionally and morphologically similar [96]. For example, CFTR, a key gene encoding an anion channel in ductal epithelial cells, is involved in the transport of chloride and bicarbonate. Currently, abnormalities in the CFTR gene are not only a cause of hereditary pancreatic cystic fibrosis [98], but are also implicated in the pathogenesis of idiopathic chronic pancreatitis [99]. Meike et al. obtained iPSCs from cystic fibrosis patients by plucking hair and constructed an organoid model for estimating whether patients were eligible for specific treatments [100]. Therapeutic approaches utilizing cmRNA within organoid models have also been reported [100]. Because CFTR mutations are not necessarily correlated with the onset of pancreatitis, PRSS1, SPINK1 and CTRC mutations may also lead to pancreatitis. Pancreatic organoid models can be built to identify possible targeted CFTR modulators for treatment, regardless of congenital or acquired CFTR insufficiency [97]. Similarly, ethanol induces mutations in CFTR that render alcoholic pancreatitis more severe. The generation of an organoid model in which CFTR damage induces a chain of events in response to calcium overload has demonstrated an increase in calcium excretion as a new therapeutic direction for alcoholic pancreatitis [101]. The proposal of organoid models, which have been used to explore new areas in pancreatitis and pancreatic cystic fibrosis, points to new therapeutic directions.

Pancreatic neuroendocrine cancer is the second most common group of pancreatic malignancies, and the lack of representative in vivo or ex vivo markers is an important cause of diagnostic difficulty, which leads to devastating mortality rates [102]. To search for potential targets, organoids can be utilized to construct suitable drug screening platforms that can aid the diagnosis and targeting of rare tumours [103]. In addition, an endocrine tumour organoid biobank has been constructed, and the link between some gene alterations, chromosome morphology and cell phenotype has been clarified, which will provide new insights into the study of rare tumours [104]. Although endocrine tumours present a low mortality and are of relatively low interest to researchers, organoid models provide the potential for in-depth investigation.

## 6. Limitations and Perspectives

Organoids are not currently a well-established model. Compared with in vivo models, the construction of suitable extracellular stroma, immune environments and vascular connections is difficult in organoids, which makes it difficult to explore the potential association of the relevant tissue with the surrounding stroma. As organoid research has gradually advanced, organoid models have become more sophisticated. Therefore, exploring the interactions between the various components and constructing models with genetic polymorphisms will shed new light on organoid research. The addition of niche factors to culture media to construct various types of organoid models has become an indispensable component of organoid research. Additionally, because organoids are mostly established in Matrigel, the components of which are not fully defined, potential unknown components could adversely affect subsequent studies with organoid models. Another problem related to pancreatic organoids is the establishment of a pure cancer organoid because the tumour mass may be contaminated by overgrown normal pancreatic cells. The current treatments are limited to the selective construction of organoids containing common genetic mutations, which limits their use to studies of rare tumour subtypes and tumour gene polymorphisms.

Even with the many current challenges, the possibility that organoid technology will positively impact basic disease research and clinical progress is undisputed. The combination of organoids with CRISPR–Cas9 gene editing technology has enabled faster construction of tumour models into targeted bodies while generating tissues that highly resemble patient cancers and enable the improved study of advanced cancers. Although the culture environment of organoids is not currently completely understood, Matrigel-free hydrogel and decellularized matrix culture protocols are gradually being refined, and in-depth progress is beginning to be made and is not confined to specific organoids. In addition, the convergence of organoids with chip technology in recent years has further optimized the culture environment of organoids. These technologies increase organoid perfusion by systematically integrating organoids with patterns of connections between surrounding stromal, immune, and endothelial cells, and allow cells to make broader contacts with specific factors, which allows the generation of organoids of superior quality to meet the demands of basic research and high-throughput drug screening. Currently, there exists a lack of indicators for the early diagnosis of pancreatic cancer; thus, by taking advantage of well-established high-throughput screening to identify potential biomarkers for the early stages of disease, organoid research will make it more difficult to target pancreatic cancer.

## Figures and Tables

**Figure 1 ijms-24-04027-f001:**
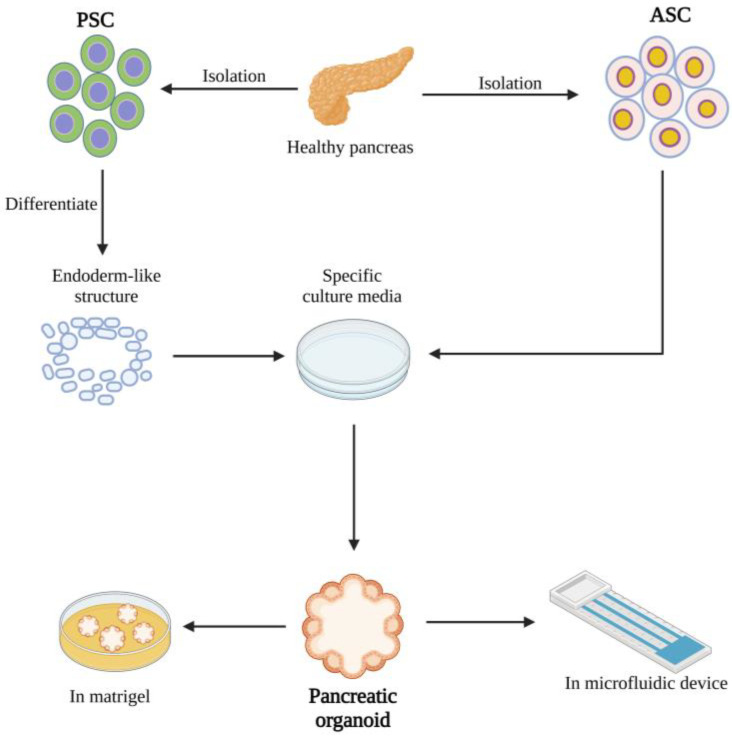
Pancreatic organoids can be constructed in a variety of forms. Pancreatic organoids can take many forms, and there are many different methods for their construction. Human PSCs (hPSCs) can be constructed by targeted differentiation using activin A and subsequent implantation into culture media containing specific growth factors. ASCs emphasize the use of specific adult stem cells as hPSCs for culture to construct organoids. In addition to conventional culture in Matrigel, organoids can be combined with gene chip technology to achieve better research results. (Created with biorender.com).

**Figure 2 ijms-24-04027-f002:**
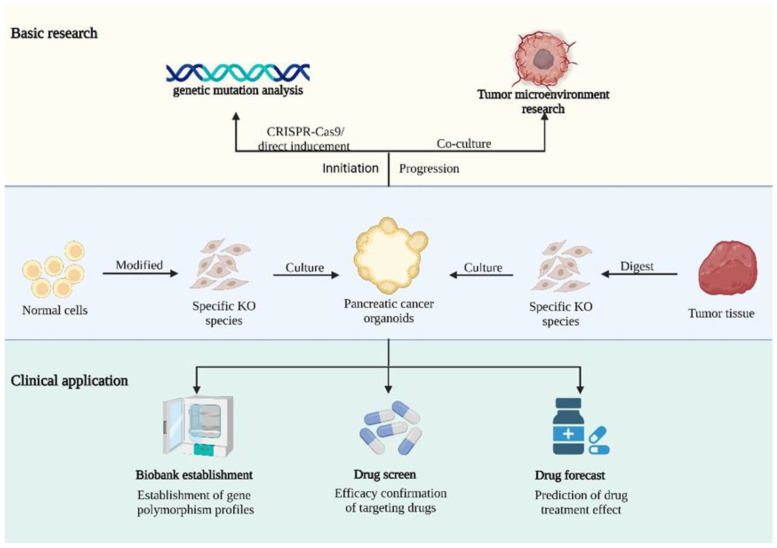
Illustration of pancreatic cancer organoid applications. Pancreatic organoids are of great importance in both basic research and clinical settings. In basic research, gene-edited specific organoids have furthered the study of cancer development and progression. In clinical applications, organoids are involved in the diagnosis and treatment of patients through the construction of biobanks, drug screening, and the evaluation of therapeutic options using a variety of strategies. (Created with biorender.com).

## Data Availability

Not applicable.

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
