# Peer review of "Pancreatic Organoids: A Frontier Method for Investigating Pancreatic-Related Diseases"

_ijms, 2023, doi:10.3390/ijms24044027_

Round 1

Reviewer 1 Report

This manuscript is well-organized and a concise summary of work in the field. However, there are some language and stylistic concerns that should be addressed before the work is accepted:

-        Figure 1: The authors have an arrow going from “induced” to the “tumor cells” that crosses over another line. This should be changed, as the aesthetics are bad. I would recommend that the authors remove the bottom part of figure 1, which shows pancreatic cancer organoid formation, as this information is shown much more clearly in figure 2. Alternately, the authors could eliminate figure 1 entirely, and insert another set of illustrations into figure 2 that concerns non-cancer cells.

-          Line 12-13: “The pancreas is a vital organ….” This sentence is a bit of a non-sequitur: how does the immobility of the pancreas within the peritoneum lead to asymptomatic conditions? I would recommend deleting this sentence entirely.

-          Line 19: “…intended for studying these corresponding diseases.” Such as?

-          Line 20: “Compared with traditional models….” Such as?

-          Line 21: “do not require ethical compliance”: The authors make such claims on more than one occasion in the review, and I don’t think that such a claim is accurate. Any animal or human-derived tissue will require some form of compliance, such as review by an animal care committee or Institutional Review Board. Now, it is reasonable to say that organoid models for pharmaceuticals require less complicated compliance procedures (and have fewer ethical concerns) compared to human or animal trials, but the cells that are used to construct the organoids have to come from someplace, and so issues of ethics and compliance still remain.

-          Line 35-36: “secretion of various hormones by nerves and bodily fluids”- I would recommend editing this to “secretion of various hormones, such as….” and make no mention of nerves or bodily fluids. As currently written, this sentence implies that the nerves and the bodily fluids are doing the secreting, rather than being affected by the secretions of the pancreas.

-          Line 46-47: “the advantages of transforming….” As written, it is not clear what the authors are trying to say. This could perhaps be rephrased as “…the ever-pervasive field, it remains difficult to translate results into human clinical applications.” The authors should also give an example or two of a case where a laboratory result did not translate into the clinic.

-          Line 49: “Although cells inevitably suffer from the obstructions of their resources….” This statement is not clear. I suggest that this sentence be edited to read, simply, “The immeasurable genetic diversity of different pancreases is not recapitulated by available cell lines [3]…”

-          Line 57: “however, they are a mixture of time and capital consumption” should be re-written as “however, establishing these lines is time and capital-intensive…”

-          Line 67: “These cysts…have the potential for stem cells…” should be rephrased, perhaps to “are still multipotent”

-          Line 114: “few patients…when suffering”: I believe the authors meant “few patients (less than 5%) survive longer than 5 years.”

-          Line 124: should be written as CRISPR-Cas9

-          Line 139-142: “Uncontrolled expression of more regulators…preclinical phase”: this sentence as written is unclear and should be re-worded.

-          Line 146: “Days before organoids were created…” this is somewhat informal, and should be revised, perhaps to “Prior to the discovery of organoids….”

-          Line 159: “specific organoids exhibit specific biochemical inclinations….” Such as? The authors should specify or provide examples.

-          Line 178: “ sub-optimal microenvironment” Sub-optimal for what? For survival of the cells? For the efficacy of chemotherapy?

-          Line 180: “that heterogeneity is found…” Could the authors be more specific? Heterogeneity of cell types? Mechanical properties of the tumor?

-          Line 181: “differences are detected in size and structure”: does this refer to differences from batch-to-batch, or within each experiment/replicate/sample?

-          Line 186 refers again to heterogeneity. Heterogeneity of what, specifically?

-          Line 187: the authors should more clearly connect CAFs to dense stroma of tumor mass. I also recommend deleting “in other words” on line 187.

-          The previous comment about line 187 could potentially be resolved by re-ordering the paragraph, such that the sentence “Dense stroma….specialized elimination [42]” now comes immediately after citation 45.

-          Line 204: the authors should consider defining what is meant by “deserted” TME.

-          Line 212-214: “Moreover, although many other factors….new therapeutic targets.” I think as currently written this is somewhat unclear, and I believe what the authors are trying to say is that certain factors are correlated with aggressive tumors, but not necessarily with tumorigenesis. To convey this sense more clearly, I think the words “shows a malignant intensity in tumor progression” should be altered to “are associated with a malignant intensity in tumor progression.”   

-          Line 218, “Basically” should be changed to “In addition” or similar.

-          Line 219: the acronym HIF-1 should be spelled out in full.

-          Line 220: USP25 should also be spelled out in full, and its function should be briefly described.

-          Line 226: “stellate cells” should be used instead of “they”

-          Line 229-230: “limitation of two types of cells….” What are these limitations?

-          Line 237-240: Are the authors suggesting the implantation of cancer organoids into patients as a potential therapeutic, as doing so would recruit the immune system and attack the tumor that is already present? The authors may want to back away from this suggestion, as, first, it seems unlikely that such a therapy would pass regulatory scrutiny, and second, such an implant might not solve the problem of cancer-caused immunosuppression that the authors note in the rest of the paragraph.

-          Line 269-272: it is unclear to me what the authors mean in this sentence.

-          Line 275: citation 64 is not Toni at al, but rather Seppala et al.

-          Line 289-292: “Upregulation of cell cycle-related…” This sentence doesn’t make any sense. Increased sensitivity to gemcitabine leads to cellular insensitivity of gemcitabine?

-          Line 299-300: could the authors provide a few examples of the insights provided by the tool used in reference 69?

-          Line 308-310: could the authors specify which basal markers, and which ductal markers?

-          Line 317: “and an increase in tumor specimens”: an increase of what? Tumor size? Tumor aggressiveness? Tumor metabolism?

-          Line 318: recommend changing “organoids, the depletion of which” to “organoids. The depletion of this complex downregulates….”

-          Line 319: “related genes and transcription factors.” What are the implications of these findings/what do they suggest?

-          Line 319-321: “Despite the tumor mass….strategies.” Recommend deleting this sentence as unnecessary.

-          Line 321: It feels like there is a missing transition sentence between discussion of the PAF1 subcomplex, and the discussion about platelets.

-          Line 331: “Pancreatic cancer remains the most favourable cancer for organoid research.” Why? What difficulties are faced by, for instance, lung organoids in recapitulating lung cancer, which is not seen in pancreatic organoids? OR, do the authors argue that pancreatic cancer is unique in not having good non-organoid models, whereas other cancers do?

-          Line 338: Why is the exclusion of Matrigel especially important in pancreatic cancer models? The authors refer to this briefly in lines 440-442, but should probably put an explanatory sentence near line 338 as well.

-          Line 387: “many markers of cell function and maturity were elevated” Such as?

-          Line 389: rather than “relative insufficiency”, I recommend “relative lack of understanding.”

-          Line 404: Perhaps use “For example” rather than “in addition”, so the sentence now reads “For example, CTFR, a key gene encoding….”

-          Line 409: “pluripotent stem cells using hair for assessing”: do the authors mean “pluripotent stem cells derived from hair” instead?

-          Line 411-413: “Since CTFR mutations are not…carry these mutations.” This sentence should probably be moved to after reference 96. The authors should then briefly discuss what sorts of non-CTFR mutations have been identified which lead to pancreatitis, and the role organoids have played in finding these mutations.

Reviewer 2 Report

The authors present a review of the literature on pancreatic organoids. It is an interesting topic, however, the manuscript is difficult to follow and requires English editing; e.g. Line 52-53: “GEMMs, which allow target gene mutations to be introduced into mice to stimulate the human pancreas and perform experiments” – I don’t understand this sentence about stimulating human pancreas by introducing mutations into mice.

 Line 151-152: „Creating organoids is relatively specific, and foetal pancreatic organoids and same genomic mouse organoids[29] implanted in mice failed to achieve a satisfactory result” – I do not understand what the authors mean by this sentence.

Several remarks are very general knowledge, not adding to the overall value of the text (e.g. the whole first paragraph in the Introduction, or in the "3.3 Drug screening and personalized medicine", the first and second paragraph partially repeat the same general information on chemotherapy. Similar information is repeated again in line 285 and 295. This should be summarized in one paragraph).

On the other hand, the authors do not elaborate on other relevant topics, e.g. the crispr-cas9 technology (line 124), gene chip and 3d bioprinting (line 392).

All abbreviations should be explained when first used.

In my opinion, the manuscript might be of interest to the readers, however, may be reconsidered for publication only after extensive edition.

Author Response

Response to reviewer 2

  1. The authors present a review of the literature on pancreatic organoids. It is an interesting topic, however, the manuscript is difficult to follow and requires English editing; e.g. Line 52-53: “GEMMs, which allow target gene mutations to be introduced into mice to stimulate the human pancreas and perform experiments” – I don’t understand this sentence about stimulating human pancreas by introducing mutations into mice.

Response: Many thanks for the comments. It should be "simulate" instead of "stimulate". There are many similarities between humans and rats, so animal experiments can predict the results of human experiments to some extent.

  1. Line 151-152: „Creating organoids is relatively specific, and foetal pancreatic organoids and same genomic mouse organoids[29] implanted in mice failed to achieve a satisfactory result” – I do not understand what the authors mean by this sentence.

Response: Thank you very much for the comment. There are some problems with the representation of the original sentence. Our intention was that there is some controversy in constructing in vitro induced tumor-like organs to be implanted in mice for cancer research. Because this has some differences with the occurrence of pancreatic cancer in the physiological situation, it can be considered artificial. We amended the relevant content to make it clear.

  1. Several remarks are very general knowledge, not adding to the overall value of the text (e.g. the whole first paragraph in the Introduction, or in the "3.3 Drug screening and personalized medicine", the first and second paragraph partially repeat the same general information on chemotherapy. Similar information is repeated again in line 285 and 295. This should be summarized in one paragraph).

Response: Thank you for your constructive suggestion. We have combined some of the repetitive content to reduce redundancy and to make a more logical presentation. In addition, some general knowledge has been removed from the introduction, leaving only a little as background to introduce the topic.

  1. On the other hand, the authors do not elaborate on other relevant topics, e.g. the crispr-cas9 technology (line 124), gene chip and 3d bioprinting (line 392).

Response: Thank you for your insightful suggestion. We have added a new paragraph in the "creating organoids" section on how new technologies can be combined with organoids to achieve better research results. CRISPR-Cas9 technology is more effective in achieving site-specific gene modifications and can induce cancer in normal pancreatic cells when combined with organoid technology to better study the pathogenesis of pancreatic cancer. Gene chips provide a more suitable culture environment for organoids because they require fewer cells, making high-throughput drug screening possible. 3D bioprinting then allows for the construction of novel media or surrounding substrates for organoids. In a well-defined culture environment, it reduces the interference of other factors in the research results.

  1. All abbreviations should be explained when first used.

Response: Many thanks for the comment. We have improved the content so that most of the abbreviations are spelled out the first time they are used, except for some gene names.

In my opinion, the manuscript might be of interest to the readers, however, may be reconsidered for publication only after extensive edition.

Response: Many thanks for your comments. The manuscript has been edited and proofread by American Journal Experts at the time of initial submission, and given the time constraints, this revision refines the content changes and is self-reviewed. In addition, American Journal Experts has agreed to re-edit this paper and we would appreciate if you would provide additional time for second revision.

Reviewer 3 Report

The manuscript need a professional proofreading editing for the various flaws present in the text (some of them underlined) in the attached file

Author Response

Response to reviewer 2

The manuscript need a professional proofreading editing for the various flaws present in the text (some of them underlined) in the attached file.

Response: Many thanks for pointing out our fault. The latest version has also been re-edited by American Journal Experts and we have re-edited the pointed sentence and proofread the whole manuscript. The manuscript now may be more understandable.

Reviewer 4 Report

The article in its current form has been significantly improved and is ready for publication

Author Response

Response to reviewer 4

The article in its current form has been significantly improved and is ready for publication

Response: Many thanks for your comments. We are very honored to hear your encouraging comments. We hope our revised version now would be able to publish in your journal.

Reviewer 5 Report

Recommendation: Minor revisions needed as noted.

 This manuscript seems to have a good theme and a well-made structure as a review paper. I think that this paper is almost finished and progressed for publication. Taking into account the quality of work and scope of the journal, I would recommend the minor revision according to the following comments.

 Comments:

 # 1. The author should check `author guideline` from this journal and modify the references.

Author Response

Response to reviewer 5

This manuscript seems to have a good theme and a well-made structure as a review paper. I think that this paper is almost finished and progressed for publication. Taking into account the quality of work and scope of the journal, I would recommend the minor revision according to the following comments.

 Comments:

 # 1. The author should check `author guideline` from this journal and modify the reference

Response: many thanks for your comments. We have checked the author guideline and applied the reference form as ACS as the journal required.

Round 2

Reviewer 1 Report

Thank you for your efforts and hard work in revising this manuscript in accordance with my suggestions. I have only a few minor comments, and recommend accepting with minor revisions:

"It is acceptable that after the abortion of TGN1412, the activator of CD28, there is no excuse for any further caution in clinical translation of basic research." - I do not understand what the authors mean in this sentence.

Line 154: "Prior to the discovery of organoids were created," can be shortened to "Prior to the discovery of organoids,"

"even 186
used in same cell lines and cocultured with fibroblasts, and slightly differences are 187
detected in size and structure in each organoid:" I suggest "cocultured with fibroblasts, and slight differences are detected in size..."

Author Response

Response to reviewer 1

  1. Thank you for your efforts and hard work in revising this manuscript in accordance with my suggestions. I have only a few minor comments, and recommend accepting with minor revisions:

Response: Many thanks for your comments. We are pretty honored to make an appropriate revision to meet your comments. The re-edit version by American Journal Experts has been uploaded. The certification of which has been uploaded on supplementary materials. Thank you again for your sincere and detailed comments!

  1. "It is acceptable that after the abortion of TGN1412, the activator of CD28, there is no excuse for any further caution in clinical translation of basic research." - I do not understand what the authors mean in this sentence.

Response: Thank you very much for the comment. In the re-edit version by American Journal Experts, the sentence has been corrected to be more accessible. Overall, there were significant differences between the performance of TGN1412 in mice and humans, and errors in the clinical implementation that ultimately caused the medical disaster. We would like to list the TGN1412 incident as a lesson of failure to conduct a critical review of clinical translation.

  1. Line 154: "Prior to the discovery of organoids were created," can be shortened to "Prior to the discovery of organoids,"

Response: Many thanks for the comments. In the re-edit version by American Journal Experts, this sentence has been corrected. We have revised it in the final version in accordance with your suggestions.

  1. "even 186 used in same cell lines and cocultured with fibroblasts, and slightly differences are 187 detected in size and structure in each organoid:" I suggest "cocultured with fibroblasts, and slight differences are detected in size..."

Response: Thank you for your constructive suggestion. In the re-edit version by American Journal Experts, this sentence has been corrected. We have revised it in the final version in accordance with your suggestions and made appropriate changes to the syntax.

Reviewer 2 Report

Dear Authors,

Thank you for your responses and corrections. The corrections are acceptable and the manuscript will be fit for publication after additional English editing (there are still some misspellings and minor errors).

Author Response

Response to reviewer 2

  1. Thank you for your responses and corrections. The corrections are acceptable and the manuscript will be fit for publication after additional English editing (there are still some misspellings and minor errors).

Response: Many thanks for the comments. As you may concerned, the re-edit version by American Journal Experts has been uploaded as the latest manuscript. The certification of which has been uploaded on supplementary materials. Thanks again for your generous suggestions and comments.